# Genes, Heritability, ‘Race’, and Intelligence: Misapprehensions and Implications

**DOI:** 10.3390/genes13020346

**Published:** 2022-02-15

**Authors:** Neil S. Greenspan

**Affiliations:** Department of Pathology, School of Medicine, Case Western Reserve University, Cleveland, OH 44106, USA; nsg@case.edu

**Keywords:** IQ, genotype, phenotype, gene-environment interaction, cultural mediation of effects attributed to allelic variation

## Abstract

The role of genetics in determining measured differences in mean IQ between putative racial groups has been a focus of intense discussion and disagreement for more than 50 years. While the last several decades of research have definitively demonstrated that genetic variation can influence measures of cognitive function, the inferences drawn by some participants in the controversy regarding the implications of these findings for racial differences in cognitive ability are highly dubious. Of equal importance, there is no compelling scientific rationale for focusing on and devoting substantial effort to determining mean differences in intelligence or other cognitive functions between groups with incompletely defined and dynamic (and therefore not definitively definable) boundaries.

## 1. Introduction

“7  Whereof one cannot speak, thereof one must be silent.”Ludwig Wittgenstein, *Tractatus Logico-Philosophicus*, p. 90, 1922

September of 2021 marked fifty years since I began my undergraduate education in, as best as I can recall, a slightly more dramatic setting than I expected: forced to traverse a group of protesters outside the building where one of my initial classes was to meet. This first-semester course was Social Sciences 15, an introduction to psychology taught by Professors Roger Brown and Richard Herrnstein, both eminent researchers in the field. Not surprisingly, given my status as a freshman, I had been completely unaware of their reputations or the research upon which those reputations rested.

The protesters were there because Professor Herrnstein had written an article entitled “I.Q.” that was published that very month (September 1971) in *The Atlantic* [1]. In this article, Herrnstein argued that American society was separating into two strata due to assortative mating: (1) rich and more intelligent people and (2) poor and less intelligent people. Unsurprisingly, Professor Herrnstein’s thesis generated a sometimes-intense controversy that still reverberates.

For example, as recently as this past September, Kathryn Paige Harden, a Professor of Psychology at the University of Texas, Austin, published a book (“The Genetic Lottery: Why DNA Matters for Social Equality”) relevant to this controversy [2]. Harden’s book directly addresses the role of genes in influencing IQ or intelligence and other issues raised by the thesis of Herrnstein in his 1971 article and further explored with Charles Murray in a widely discussed and highly controversial book (“The Bell Curve: Intelligence and Class Structure in American Life,” 1994) [3].

As an aside, I will note that I had reason to meet with Richard Herrnstein in person at the beginning of the second semester due to a family matter that required my leaving the campus for a week. Compared to some other professors I had at that time, Professor Herrnstein was relatively understanding and empathetic. The point of my recounting this interaction is only to note that my impression of Professor Herrnstein on a personal level was positive. 

A critical element in Herrnstein’s argument about the inevitable stratification of society was the claim that the heritability of I.Q. was high, about 80%. My purpose here is not to address the entire argument Herrnstein made in his magazine article of 1971 or the subsequent version of that thesis put forward by Herrnstein and Murray in their joint book. Instead, I want to focus on the roles of genes in shaping cognitive function and the concept of **heritability**, and what it does and does not mean. In my view, understanding heritability is an issue of central importance in the controversy regarding genes, cognitive ability, and race and a matter of pervasive misunderstanding in the discussion of the role of genes in human behavior.

Two years after my introduction to psychology, my interest in this topic was further stimulated in part by another professor I had for an introductory course on population and evolutionary genetics. This course was taught by Richard Lewontin, who died, as of the time of this writing, just six months ago in early July of 2021. Professor Lewontin was justly celebrated as among the most critical population and evolutionary geneticists of the second half of the 20th century [4,5]. He was known for broad interests, lucid writing for both fellow scientists and general audiences, and remarkably penetrating insights. 

Professor Lewontin also happened to have a strong interest in the complexities that characterize genotype–phenotype relationships [e.g., see [6] for a relatively non-technical and accessible account of this topic] and how these relationships are captured by heritability, which makes sense because one of the two technical definitions of the term, so-called narrow-sense heritability, is relevant to evolution and to practical applications of selection for traits in plants and animals that are of economic value in agriculture. Therefore, in his course on population and evolutionary genetics, he devoted a significant amount of time and effort to exploring the subtleties of heritability with his students. Unfortunately, many commentators who refer to this concept in arguments about various topics involving human behavior or brain function seem not to understand what heritability magnitudes mean or fully imply.

## 2. Examples of How Heritability Is Misunderstood

Some months ago, in the latter half of 2021, I listened to a podcast (“The Good Fight”) hosted by Yascha Mounk, a professor of international affairs at Johns Hopkins, in which he interviewed Arthur Brooks [7], currently a member of the faculty at the Harvard Kennedy School of Government. Harvard has faculty members in several schools and departments who study aspects of genetics and have contributed significantly to our understanding of gene structure, gene function, gene replication, the transmission of genes, and how genes relate to evolution. However, as an institution focused on politics and policy, the Kennedy School is not the locus of a great deal of high-quality genetic or other biological research.

In the podcast, Brooks stated, with apparent self-confidence, that happiness was 48% “genetic,” by which he probably meant to say that happiness was 48% heritable. He “helpfully” added for the sake of the listeners that “the other half is circumstantial”. Presumably, Brooks meant that the other half of happiness depends on environmental factors, not genes.

Although these statements may appear to be meaningful, they are actually challenging to take seriously as scientific claims. Perhaps they even merit the famous riposte of the quantum physicist, Wolfgang Pauli, to a colleague’s proposal that “It’s not even wrong.” For instance, it is not clear what Brooks means by “genetic” or by “the other half [of happiness]”.

Even a quote from a highly regarded economist regarding the relative importance of “nature” vs. “nurture” in determining the traits of progeny contains a number of doubtful assumptions about heritability and the factors that influence phenotypes [8].

“I asked Justin Wolfers, father of Matilda, for his take on the power of nature versus nurture.”


*“Justin WOLFERS: Uh, who cares? You do the best you got with what you got. So if it’s 80 percent nature, it still leaves me with 20 percent. If it’s 20 percent nature, it leaves me with 80 percent. And either way, I want to get that part of the puzzle right”.*


While I would agree with Professor Wolfers that as a parent, you do the best you can to raise your children irrespective of heritability values, his comment seems to imply or at least might be interpreted to mean that heritability values can be meaningfully applied to individuals, that “nurture” corresponds entirely to “what parents do,” and that “nature” is solely about what genes are inherited from parents. None of these claims are completely accurate.

A third example comes from an exchange between Ezra Klein and Sam Harris. [9]. Klein is a well-known journalist who is highly critical of the camp advocating for the importance of studying the heritability of IQ in determining social policy. Sam Harris, an author, and intellectual who is seen as defending those advocating for the study of the heritability of IQ, which people such as Charles Murray believe can be useful in determining social policy. At one point in a rather lengthy exchange with Klein, Dr. Harris, who has a Ph.D. in neuroscience, states that

“We don’t know of an environmental intervention that reliably changes people’s IQ. Murray is right about that”.

I believe that Harris is suggesting the non-existence of environmental interventions that reliably boost IQ scores. If so, perhaps he is, strictly speaking, correct on that point, meaning that IQ scores cannot be increased through social programs. However, I will note that James Heckman, an economist at the University of Chicago, a Nobel laureate, and an extremely eminent investigator of the outcomes attributable to social interventions, has provided strong evidence for the ability of some interventions to increase life success in several respects in ways that some might have assumed would require greater cognitive ability [10]. I believe it is worth directly quoting from Professor Heckman’s website about his research project on the Perry Preschool:

“While Perry failed to permanently increase a crude IQ measure of the treated, simplistic measures of cognitive achievement prove to be poor indicators of life success. Children treated with early childhood education have significantly better life outcomes than untreated children. Treatment in Perry significantly increased the participants’ employment, health, cognitive and socioemotional skills and reduced the male participants’ criminal activity, especially violent crime. Improvements in childhood home environments and parental attachment are seen as an important source of the long-term benefits of the program”.

I also have to note that there is strong evidence that exposure to various environmental influences can reliably decrease effective IQ. For example, a meta-analysis of studies on environmental lead (abbreviated as Pb by chemists) exposure and IQ published this year presents overall results from seven reports that met the inclusion criteria for the new assessment [11]. The authors of the meta-analysis conclude as follows: 

“The full-scale IQ score[s] are inversely proportional to the blood Pb values…”

My thesis addresses seven critical and common misunderstandings:(1)The evidence is overwhelming that genetic variation at many genomic loci substantially influences variation in intelligence and other cognitive capacities or behaviors. Some of this impact is mediated directly by biochemical mechanisms involving the gene products specified by the implicated genes. Still, some of these influences may arise through interactions between genetic and environmental factors and be, at least in part, culturally mediated.(2)In spite of this first point, the heritability of a phenotype within a population does not determine the extent to which genetic variation is responsible for mean differences between populations that can be differentiated from one another with respect to factors that exert significant influence on variation in a phenotype of interest. Furthermore, the genetic variation that influences variation in measures of cognitive performance within a population is not necessarily identical to the genetic variation that influences variation in mean measures of cognitive performance between populations.(3)Related to the preceding point, accepting a role, even an important role, for genetic variation in influencing variation in intelligence and other cognitive capacities or behaviors in no way necessitates concluding that individuals with what are defined as lesser apparent capabilities cannot be helped by changes in environmental factors.(4)Evidence of genetic influence on mean measures of cognitive performance between putative racial groups does not necessarily imply that the relevant genes are directly involved in determining brain functioning.(5)The implicit assumption of some participants in the controversy that genes involved in influencing IQ in some putative racial groups are clearly and unalterably superior to those same genes in other racial groups, creating a static hierarchy, is unjustified.(6)Although there may be no convincing evidence that intentional interventions can directly raise putatively inherent IQ, there are clearly environmental factors that can decrease putatively inherent IQ. Therefore, eliminating or reducing the impact of such factors, especially early in life, can potentially increase IQ relative to what it might have been if those negative influences were not eliminated or attenuated.(7)There is no compelling scientific reason to study the mean values for measures of cognitive or behavioral phenotypes associated with putative racial groups because individuals should be evaluated as individuals.

## 3. What Is Heritability?

What then is heritability and how does heritability differ from “being genetic?” One can also ask what it means to claim that a phenotype is heritable to any given extent?

Heritability is a measure of the extent to which phenotypic variation, i.e., variation in a well-defined trait or characteristic (phenotype) of an organism, is attributable to genotypic variation [12,13]. Genetic variation is the existence at particular locations in the genome (loci; singular: locus) of different nucleotide sequences (alleles; singular: allele). A high or low value for the heritability of a trait in no way informs us about whether a trait is “genetic” in the sense that genes are substantially involved in the developmental process by which a trait in a given individual is realized. 

For example, the number of fingers on human hands, which is five in the vast majority of cases, is highly dependent on the functions of proteins that are encoded by human genes, but much of the variation in that number results from accidents involving power tools or machines, i.e., environmental influences. The variation in the lengths and number of fingers on human hands that results from genetic variation, and there is such variation [14,15], has helped to identify the genes that are involved in constructing human hands in development through the stages from embryo to fetus to neonate. A related point is relevant to variation in cognition.

If happiness, returning to the concern of Professor Brooks, as defined (measured) in some particular way, were 48% heritable, that would mean that 48% of the variation in happiness among the members of a population was attributable to variation in the alleles possessed by different individuals at one or more genetic loci in that population. Key points to note are:(1)The value of the heritability for a given phenotype is a property of a particular population and not of any single individual.(2)The extent of the impact of allelic variation at a particular locus on variation in a trait, such as happiness, cognitive function, height, eye color, or skin color may either correlate or interact with (i.e., depend on) one or more environmental factors such that disentangling the relative importance of genetic versus environmental factors for individuals becomes highly challenging or practically impossible.(3)So, the heritability value for a trait might be expected to vary depending on the environmental factors operative within a population and the distributions of the impacts of those environmental factors on the members of a population, as well as the precise manner by which the trait is assessed or, if quantitative, measured.(4)Heritability can vary for a trait depending on when in the developmental process (e.g., neonate vs. child vs. adult) the traits are assessed in members of a population.(5)The magnitude of heritability for a given trait is influenced by the particular allele frequency distributions at the relevant loci in the population studied and for which other populations may differ substantially.(6)As implied by the preceding points, changes in the relevant allele frequency distributions (which are not merely possible but over sufficiently long periods of time likely) or in the presence or distribution of environmental factors can lead to alterations in the degree of heritability.(7)Combining some of the above points, the value of the heritability is not an intrinsic property of a gene, a trait, or a gene-trait pair for a given population or whole species.(8)It is not generally possible to delineate all relevant environmental factors or assess how extensively they affect all members of a population.(9)As noted above, even a very high percentage of heritability for a specific phenotype has **no** implications for how amenable to change that phenotype will be if exposed to altered environmental inputs, such as medical or social interventions that were not previously present in the study population.(10)Therefore, determining a high heritability for a trait does not automatically argue against actions in social policy, medical, or other practical contexts whereby environmental factors would be modulated.(11)The influence of genetic variation on a trait can be indirect and be mediated almost completely by factors other than inherent biochemical mechanisms such as culturally grounded practices and/or beliefs such that what may be taken to be a genetic effect may be as or more appropriately viewed as originating, at least in part, from environmental causes, and(12)The vast majority of traits in the vast majority of people are the result of both genetic and environmental factors [16].

The following two sections (Section 4 and Section 5) are intended to provide context for and concrete examples relating to the above list of key points about heritability. Section 6 contains additional information about the distribution of human genetic information and the limitations of conventional accounts of race.

## 4. Genetic Fundamentals

A few basic aspects of gene function are also important to know. First, genes encode proteins and RNA molecules (gene products) that are frequently involved in two or more biochemical pathways that can differ in different types of cells and that can influence multiple traits. As a result of such multiplicity of gene product interactions, a mutation in a gene can have multiple phenotypic effects—a property referred to as pleiotropy. Second, the functional consequences of mutations at one locus can depend on the identities of alleles at other loci, a functional interaction between genes referred to as epistasis. Third, due to epistasis and gene-environment interactions and other factors, the phenotypic implications of possession of a particular allele may vary significantly in different individuals, referred to as variable penetrance, the likelihood given possession of a specific allele that an associated trait is observed, or expressivity, the extent to which the trait associated with the gene in question is expressed [17,18].

Another basic reality beyond dispute is that genetic variation affects variation in brain development and neural and cognitive functioning. A large body of experimental literature demonstrates causal connections between a variety of genetic variants of multiple types and intellectual disability. Mutations affecting the functioning of products produced by more than 100 genes on the X-chromosome alone have been associated with intellectual disability [19].

The genetic loci involved in these cases have been demonstrated in other studies to encode proteins that participate in such processes as the formation of synapses (subcellular structures that mediate communication between neurons), synaptic plasticity, neuronal migration during brain development, chromatin remodeling, which influences the amount of gene transcription (mRNA production) and ultimately the amount of gene product translation (protein synthesis guided by the mRNA), and protein half-life in the cell. Many readers will probably note that some of the processes affected by mutations of genes influencing intellectual functioning are sufficiently general in nature to have effects on a variety of traits not obviously related to cognitive function. Genes encode gene products and are not best conceptualized as each corresponding to a single trait, highlighting the centrality of pleiotropy and epistasis to understanding the complexity of how genotypes relate to phenotypes.

## 5. Genotype-Phenotype Correlations Mediated Indirectly

Over twenty years ago, I came across an instructive example of how allelic variation can influence one or more traits through indirect mechanisms that can upend one’s intuition of what genetic influence means. The article, by Donald G. McNeil, Jr., appeared in the *New York Times* in February of 1997 [20]. It was not primarily about genetics, although it did mention genes. The central focus of the piece was the social situations of individuals in Zimbabwe who were affected by albinism, from the Latin root *albus* meaning “white”.

Albino people with the most common form of the condition, oculocutaneous albinism, have reduced or absent melanin in skin, eyes and hair, all of which are tissues derived from the embryonic tissue layer known as ectoderm. Mutations in multiple different genes (i.e., genes at different genomic loci) can result in varying degrees of albinism [21].

Although the loss of melanin in skin has no obvious or direct connection to cognitive function, the reduction of melanin in the eye can result in reduced visual acuity. In Zimbabwe, where, at least as of 1997, children may not uniformly have had ready access to spectacles to correct myopia, the decreased visual acuity associated with albinism could have reduced the ability of some affected individuals to decipher writing on the blackboard in school and otherwise undercut their efforts at learning.

Beyond the effects on vision, cultural beliefs about albinos in Zimbabwe resulted in severe discrimination and even persecution. Some albino children were not even permitted to go to school because sending children to school entailed expense. Parents with limited financial resources believed their albino children would not live long enough to benefit from such education. 

Of course, many of these culturally based effects were ultimately elicited mostly because of the effects of genes that have no known direct effect on central nervous system function (see Figure 1 and legend for an example in which genetic variation affects a plant phenotype by indirect means). Consequently, a biological change caused by genetic variation that has no intrinsic connection to cognitive ability can be connected to outcomes, such as educational attainment or employment in well-paid jobs, which are presumably influenced by cognitive function (but not only cognitive function) in most circumstances. Furthermore, in this particular social and cultural context, this same literally superficial trait can lead to a reduced likelihood of getting married and having children.

In a different social and cultural environment, these same individuals with less than the usual amount of melanin in skin, eyes, and hair, might do just as well on average as those with typical amounts of melanin. For example, by undertaking simple interventions such as supplying glasses to albino children early in life and eliminating discrimination, albino children on average would likely learn comparably to other children, and albino adults could potentially achieve employment and reproductive success comparable to other adults.

The key takeaway lesson is that what appears to be a genetically determined outcome is, in fact, due to interactions between a gene or genes and aspects of the environment. Furthermore, the existence of a genetic cause for an outcome does not mean that environmental manipulations are necessarily unproductive, nor does it mean that there is a clear path to effective intervention.

In the American context and in light of the pertinent history, it should be possible to appreciate how genetic variation affecting skin color might have an impact on outcomes presumed to depend in part on cognitive capability. It is highly notable that in this setting, the presence of more melanin, not less, is associated with what some perceive as less “inherent” cognitive ability. As in Zimbabwe, however, the correlation between allelic variation at loci that encode proteins relevant to melanin production and other traits is likely mediated primarily by cultural beliefs, not the mechanisms of biochemistry and cell biology. Such genetic influence cannot, therefore, be reasonably interpreted as proof that one group of people is cognitively superior to another.

Another critical element supporting the preceding inference is that the range of environmental factors influencing relevant measures of performance thought to depend on brain function (e.g., educational attainment, employment, and professional achievement) is extremely difficult to fully delineate or assess in terms of quantitative significance. Consider the relative frequency with which children of less affluent and otherwise disadvantaged families are exposed to lead in drinking water or old paint flaking off walls, organic chemicals known to be toxic and many more of unknown impact on neural development, less optimal diets, less attentive medical care, and extreme stress including chronic violence.

In this context, it is of relevance that Patrick Sharkey has published several studies demonstrating that violence in a neighborhood can depress measures of cognitive ability for school-aged (5–17 years old) children in that community [22,23]. In other words, it is extremely unlikely that it is currently possible to accurately determine the extent to which interactions between genetic variation that has no direct impact on the brain and environmental variation affect putative group differences in measures of brain function.

Another recent publication offers evidence of the potential for stressful experiences early in life to affect future behavior by influencing the rates of gene transcription and translation in brain cells [24]. Much more research will be required to establish the potential for such mechanisms to influence measures of intelligence, but the results already reported suggest that it can be extremely difficult to disentangle effects due to genetic variations (differences in nucleotide sequence) from effects that depend on mechanisms independent of variations in nucleotide sequence. These findings also raise the possibility that individuals with particular genotypes at one or more loci are variably susceptible to the negative effects of stressful experiences early in life on later cognitive capabilities.

## 6. The Distribution of Human Genetic Variation

There is another problem with the supposition that knowing the heritability of group differences is information that can usefully guide educational or broader social policy formulation. The groups defined along standard racial lines are to a major extent arbitrary and the boundaries are necessarily imprecise and dynamic. Human evolution has not ceased.

It is clear that human genetic variation exhibits patterns that are correlated with variation in the predominant geographic origins of the ancestors of people alive today. However, dividing the people of the world into clear, discrete racial groups is misleading because the variation across the globe is most appropriately regarded as roughly continuous, as would be anticipated from an evolved and evolving lineage of organisms that can move around.

For example, there is more genetic variation among the people of Africa than among the people living on all other continents [25]. So, lumping together all Africans as one race is scientifically questionable. Adding to that the inevitability of gene flow between populations in different regions and the possibility that the same mutations can occur by chance in genomes in different locations, the idea of utterly separate and fixed racial groupings is not sensible.

Perhaps more importantly, individual students, just like individual patients or employees, need to be evaluated as individuals and not as representatives of some group. I see no value in knowing group means or comparing group means when the distributions of almost any trait of functional relevance to society will extend relatively far from the mean in both directions and be largely overlapping for these putative groups.

For example, I have known and worked with physicians and researchers originating from countries all over the world. What matters to me in these collaborative projects or tasks is the individual insights, efforts, and capacities of these collaborators, not the supposed averages for their respective racial, ethnic, or religious groups. Any professionals who prioritize the latter over the former are handicapping their own collaborations.

In 2017, John McWhorter directly addressed a central question confronting individuals who devote time and effort to assessing differences in mean IQ between large groups of people that they claim correspond to different races [26]. Why do they believe this question is worthy of investigation?

I have never seen a satisfactory answer as to why individuals who are prominent in discussions of putative racial differences in intelligence, such as Charles Murray, focus on mean group differences in measures of intelligence if the goal is policy formulation. Of equal importance, if defenders of research into the heritability of IQ or other measures of intelligence in different racial groups maintain that high heritability in a trait implies the impossibility of influencing that trait because “it is genetic” they are, in the general case, wrong.

Consider the rationale behind the Human Genome Project (HGP). The claim that was aggressively made by the most vocal proponents of the HGP was that genes are critically important in conferring susceptibility to a huge variety of diseases and medical conditions, i.e., phenotypes. More importantly, they maintained, knowing the identities of a particular patient’s genes would guide the selection of customized therapies. It was implicitly assumed that a clinical trait that was highly heritable would still likely be modifiable by medical treatment. Although the ease of identifying or producing such interventions was arguably overstated by a number of HGP boosters [27], the basic premise is reasonable: as noted above, even highly heritable traits can, in principle, be modified by manipulations of judiciously selected environmental factors.

A genetic concept that can help make sense of these claims is the norm of reaction [6]. For a given characteristic, a norm of reaction shows how the phenotype associated with a given genotype changes as a function of the environment. Careful studies done of many phenotypes in a variety of organisms in different environments reveal that these relationships can be variable in comparing one genotype to another.

For example, in investigating a phenotype of interest in agricultural plants or animals, a genotype, we can call A, that in comparison to genotype B generates a more productive phenotype in environment no. 1 might generate a less productive phenotype than B in environment no. 2. The same kinds of relationships can be found in evolutionary contexts where the phenotypes are assessed by measures of fitness and the seemingly fitter genotype in one environment is the less fit genotype in a different environment. Thus, the implicit assumption made by some participants in the arguments about genes, race, and intelligence that there is a clear and absolute hierarchy of genotypes in terms of the cognitive abilities they can produce is not justified in the absence of data that I have never seen.

## 7. Conclusions

In closing, I would like to offer a paraphrase of the epigraph at the beginning of this article: whereof one cannot speak with insight, thereof one should consider either making the effort necessary to comprehend the relevant subject matter or remaining silent. A reasonable prerequisite for usefully discussing the implications of the genetics of cognitive ability for social policy is a solid grasp of what genes are, the frequently intricate and complex mechanisms by which they influence traits via functional interactions with one another and with environmental factors, the precise technical meanings of “heritability,” and the usefulness of measurements of heritability. From what I have seen of this controversy in recent years, it would be more productive if it were more fully informed by the key points enumerated and characterized above.

## Figures and Tables

**Figure 1 genes-13-00346-f001:**
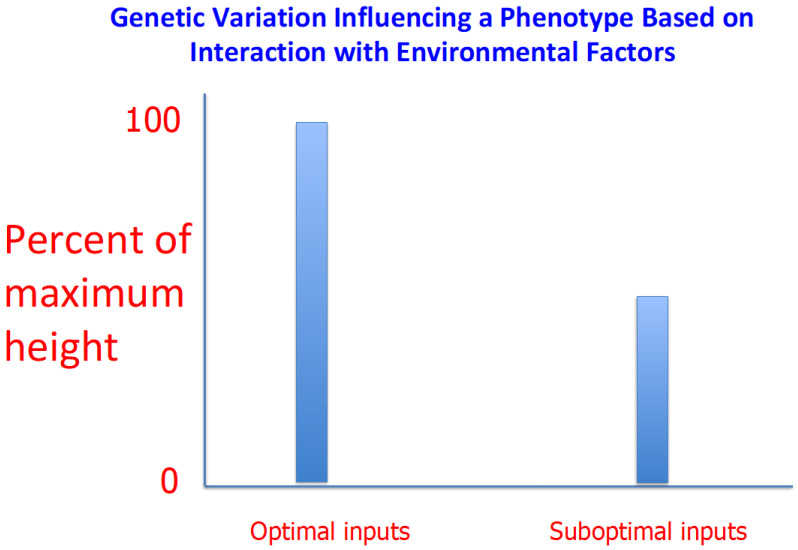
Differential growth of two populations of hypothetical flowering plants is attributable to genetic variation at one and only one locus that directly influences flower color but does not influence overall plant growth by any biochemical mechanisms. The gardener decided to provide optimal amounts of sunlight, water, and food for plants with pastel blue flowers (**left**) but not for plants (**right**) with dark blue flowers. Thus, the genetic difference between the two plants substantially influences growth in this set of circumstances but would not be expected to have any influence in other conceivable circumstances.

## Data Availability

Not applicable.

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
