# Peer review of "Genes, Heritability, ‘Race’, and Intelligence: Misapprehensions and Implications"

_genes, 2022, doi:10.3390/genes13020346_

Round 1

Reviewer 1 Report

The manuscript by Neil S. Greenspan describes the multitude of factors influencing complex traits like intelligence. It elegantly addresses the simplified view on the involvement of genes and heritability in traits. I loved the example of the human hand in lines 100-106. Although the manuscript is written primarily from a human perspective, the author decided to submit this to the section Animal Genetics and Genomics. It however nicely fits the whole nature vs nurture debate.

The manuscript is a well-written, easy-to-read story, however, being a non-native English speaker complicates checking for possible grammatical errors. I think I have found two small ones:

Line 261: should “can” be chanced in “can be”

Line 277: should “professionals” be changed to “professional”

Figure 1 does the trick, but not more than that. It is a bit of a childish drawing. From an aesthetic perspective, there is some room for improvement.

Author Response

Author responses to Reviewer 1:

I thank the reviewer for his/her constructive comments. My responses are provided below. Additional or revised text in the manuscript is indicated in red text (file with name ending in nrrt, i.e. new and revised red text). A version of the manuscript with all text in black (clean) is also uploaded.

Reviewer 1

Line 261: should “can” be chanced in “can be”

This suggested change has been implemented.

Line 277: should “professionals” be changed to “professional”

This suggested change was judged to be unnecessary.

Figure 1 does the trick, but not more than that. It is a bit of a childish drawing. From an aesthetic perspective, there is some room for improvement.

I have changed the figure into a bar graph that I hope will satisfy the reviewer.

Reviewer 2 Report

This commentary discusses several misconceptions about heritability that pervade arguments about racial differences in intelligence. The arguments the author makes are sound, and the personal experiences he shares are interesting and engaging. Although these misconceptions have been discussed elsewhere, their persistence in both scientific arguments and lay interpretations of genetic research justifies restatement.

My main suggestion for improvement is for the author to more clearly articulate/enumerate the key misconceptions he wishes to correct. In particular, in the context of explaining the heritability of happiness, the author lists 12 properties of heritability, some of which are clearly directed at correcting common misconceptions. For example, noting that heritability is a statistic for a given population at a given time, with no implications for the effect of an experimental intervention, has clear implications for common misconceptions of genetic determinism and genetic essentialism. These points might be more effective if the author initially articulated the key misconceptions that underlie arguments about race differences in intelligence and other traits. Then the relevance of these points (and the logical fallacies in their arguments), would be even clearer. (For that matter, although the example of heritability of happiness is interesting and accessible, it is a little off topic for the goal of the commentary--cognitive ability. Perhaps after describing this example, the author could mention examples of similar statements about intelligence.)

One of the points that the author mentions, particularly in the abstract and conclusion, is that race is “incompletely defined and dynamic (and therefore not definitively definable) boundaries.” This point is only touched on late in the commentary, and not explained: “The groups defined along standard racial lines are to a major extent arbitrary and the boundaries are necessarily imprecise and dynamic. Human evolution has not ceased” (lines 266-267). Given the prominence of its inclusion in the abstract, I think this point could use some elaboration.

Section 4, concerning the mediation of genetic effects through environment, is important, but also related to many of the enumerated points presented earlier. It may be that the intention of these later sections 3 and 4 are to unpack some of these points made earlier, but if so, that was not completely clear to me. 

Finally, the commentary is somewhat unfocused in that it mentions a number of issues with the whole endeavor of investigating race differences, beyond those to do with misconceptions of heritability. Although I agree with the points made (e.g., questioning why the issue is worth investigating), it may be useful to organize these points more systematically. That is, some parts of the commentary addresses misunderstanding of heritability and genetics, other parts are more philosophical and ethical. It’s true that the arguments about genetic group differences are wrong on so many levels, but without acknowledging these different levels, the commentary runs the risk of muddying its arguments.

Author Response

Author response to Reviewer 2:

I thank the reviewer for his/her constructive comments. My responses are provided below in red text. Additional or revised text in the manuscript is indicated in red text (file with name ending in nrrt, i.e. new and revised red text). A version of the manuscript with all text in black (clean) is also uploaded.

This commentary discusses several misconceptions about heritability that pervade arguments about racial differences in intelligence. The arguments the author makes are sound, and the personal experiences he shares are interesting and engaging. Although these misconceptions have been discussed elsewhere, their persistence in both scientific arguments and lay interpretations of genetic research justifies restatement.

My main suggestion for improvement is for the author to more clearly articulate/enumerate the key misconceptions he wishes to correct. In particular, in the context of explaining the heritability of happiness, the author lists 12 properties of heritability, some of which are clearly directed at correcting common misconceptions. For example, noting that heritability is a statistic for a given population at a given time, with no implications for the effect of an experimental intervention, has clear implications for common misconceptions of genetic determinism and genetic essentialism. These points might be more effective if the author initially articulated the key misconceptions that underlie arguments about race differences in intelligence and other traits. Then the relevance of these points (and the logical fallacies in their arguments), would be even clearer. (For that matter, although the example of heritability of happiness is interesting and accessible, it is a little off topic for the goal of the commentary--cognitive ability. Perhaps after describing this example, the author could mention examples of similar statements about intelligence.)

I have now provided a list of seven key misconceptions in section 2, pp. 5-6. In addition, I now provide two additional examples of misunderstandings regarding the implications of heritability assessments that are more directly related to intelligence or cognitive function in section 2, pp. 3-4.

One of the points that the author mentions, particularly in the abstract and conclusion, is that race is “incompletely defined and dynamic (and therefore not definitively definable) boundaries.” This point is only touched on late in the commentary, and not explained: “The groups defined along standard racial lines are to a major extent arbitrary and the boundaries are necessarily imprecise and dynamic. Human evolution has not ceased” (lines 266-267). Given the prominence of its inclusion in the abstract, I think this point could use some elaboration.

I have added two paragraphs on the limitations of standard conceptions of race to new section 6 on p. 11.

Section 4, concerning the mediation of genetic effects through environment, is important, but also related to many of the enumerated points presented earlier. It may be that the intention of these later sections 3 and 4 (now sections 4 and 5) are to unpack some of these points made earlier, but if so, that was not completely clear to me. 

I have added a paragraph at the end of new section 3 at the top of p. 8. to explicitly note that the following three sections give the reader illustrations of how the points about heritability are relevant in particular cases and to elaborate briefly about the nature of human genetic variation as it relates to some common ideas about human races.

Finally, the commentary is somewhat unfocused in that it mentions a number of issues with the whole endeavor of investigating race differences, beyond those to do with misconceptions of heritability. Although I agree with the points made (e.g., questioning why the issue is worth investigating), it may be useful to organize these points more systematically. That is, some parts of the commentary addresses misunderstanding of heritability and genetics, other parts are more philosophical and ethical. It’s true that the arguments about genetic group differences are wrong on so many levels, but without acknowledging these different levels, the commentary runs the risk of muddying its arguments.

I was not sure specifically how to address this concern, but I hope most readers will come away with the key insights I intend to convey.